# People Make the Difference: An Explorative Study on the Relationship between Organizational Practices, Employees' Resources, and Organizational Behavior Enhancing the Psychology of Sustainability and Sustainable Development

**Amelia Manuti** *[ID] **and Maria Luisa Giancaspro**

Department of Education, Psychology, Communication-University of Bari, 70121 Bari, Italy;
mlgiancaspro@gmail.com
* Correspondence: amelia.manuti@uniba.it

**Abstract:** The most recent developments in the field of sustainability science and the emergence of a psychology of sustainability and sustainable development have contributed to collect evidences about the fact that modern organizations need healthy and motivated employees to survive and to prosper within this fast-moving scenario. In this vein, a confirmation to these evidences came from the abundant research on *HEalthy and Resilient Organizations* (HERO), showing that when organizations make systematic, planned, and proactive efforts to improve employees' subjective resources then organizational processes and outcomes benefit in turn. Moving forward from these premises, the present study aimed to explore these assumptions within the context of small and medium enterprises (SMEs), investigating the relationships among the organizational practices, employees' subjective resources, and organizational behaviors. Two hundred and thirty-six participants working in SMEs located in the south of Italy took part. They were invited to fill in a questionnaire investigating their perception of organizational resources and practices (autonomy, leadership, communication, organizational mindfulness, and commitment to resilience), of their individual resources (work engagement and psychological capital), and finally, of some organizational outcomes (extra-role behavior). Results showed that psychological capital was a significant mediator of the relationship between employees' perception of the organizational resources and practices and extra-role behaviors. Concrete implications of these conclusions in terms of human resource management (HRM) are discussed together with limitations of the study and future developments.

**Keywords:** psychological capital; HRM practices; organizational behavior; sustainable development

## 1. Introduction

The rapid cultural, social, and economic changes that have recently invested the labor market, as well as the challenges brought about by global competition, are pushing organizations to rethink their strategies, processes, and practices, and to consider human capital as the main intangible asset that could concretely make a difference.

These evidences have been further remarked on in the psychology of sustainability and sustainable development that is an emerging transdisciplinary research area in the field of sustainability science aiming at integrating psychology within the analysis of sustainable development processes [1–3]. In this perspective, the psychology of sustainability and sustainable development focuses attention on the conditions that might enhance workers' well-being and quality of life maintaining that the paradigm of

sustainability could be extended from the ecological and socio-economic domain to the psycho–social context of working, thus, contributing to creating conditions that might promote individual as well as organizational development [1–5]. As a result, the enhancement of human resources, the management of talents, and the development of human potential are becoming good practices for the management of people inside the organizations as well as precious strategies to secure organizational survival in a turbulent market [6–8].

The main underlying assumption guiding this stream of research and consequently, good human resource management (HRM) practice in this field is that sustainability, far from being simply a matter of environmental conditions and/or of organizational processes, is strictly linked to the consideration of the unique and distinctive value that each human resource means for organizations. Therefore, "healthy people make healthy organizations" [9–12]: If organizations choose to invest in human resource management practices that are aimed at establishing and cultivating a positive P/O (Person/Organization) fit based on mutual trust and acknowledgement, then this effort will probably result in a competitive advantage for them, as those employees who are more engaged and satisfied with their job will be more likely to be more productive and efficient [13–16].

The core of the psychology of sustainability and sustainable development approach is quite attuned with another quite recent stream of research within psychology, namely Positive Organizational Behavior (POB). This perspective studies the subjective and contextual features that enable individuals and communities to thrive, presenting a real competitive advantage for organizations [12,17]. Accordingly, this positive approach allows psychology to meet management and business, thus, concretely helping organizations to create sustainable performance [18]. In an attempt to further enlarge the paradigm, this management-driven view has been enriched by a more employee-centered view of organizational performance, arguing that the mission of POB must also focus on goals, such as employee happiness and health [19], suggesting the development of a positive business value model of employee well-being, where a win–win situation for both the organization and its employees can be drawn [20].

In this vein, this approach focuses on peak performance in organizations, examining the conditions under which employees thrive and feel good at work, thus, working more and better. Accordingly, empirical research in the field of Positive Organizational Behavior [11] is giving a precious contribution to HR practice by studying "what is positive, flourishing, and life-giving in organizations. Positive refers to the elevating processes and outcomes in organizations. Organizational refers to the interpersonal and structural dynamics activated in and through organizations, specifically considering the context in which positive phenomena occur" (p. 731 [21]).

Within the wider framework of POB, Healthy and Resilient Organizations (HERO) constitute a key element. HERO are those organizations that, espousing this people-based view of management, engage actively and systematically to improve employees', teams', and organizational processes and outcomes, enhancing sustainable development [22,23]. HERO are also "resilient" because they know how to manage change, how to adjust to challenging and unpredictable situations, how to maintain desirable outcomes [18,24–32].

HERO organizations can be distinguished from more traditional, and we may say "toxic" organizations, because of three specific features.

First, because they constantly strive to implement "healthy organizational resources and practices", aimed at improving the work environment at the task (e.g., autonomy), interpersonal (e.g., transformational leadership styles), and organizational (e.g., Human Resources—HR—practices) levels, especially during times of turbulence and change [15,33–35]. Second, because, given their "healthy" approach to HRM, they have "healthy employees/teams" (e.g., trust, work engagement), who show high levels of psychosocial well-being. Finally, because of the positive interaction between the previous features, they show "healthy organizational outcomes" (e.g., high performance, corporate social responsibility, reputation).

Therefore, in view of these reflections, the authors have developed a model according to which "healthy" practices and "healthy" employees' can have an influence on "healthy" organizational outcomes (Figure 1). Consequently, to work on people, to make them the pivot of the organizational system will be a significant advantage for organizations striving to improve their performance and wishing to stay sustainable and competitive through time.

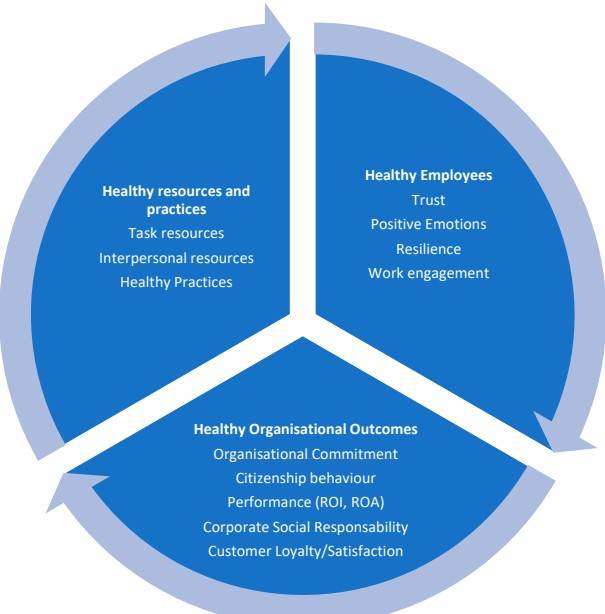

**Figure 1.** The HERO Model—HEalthy and Resilient Organizations (adapted from 23).

Inspired by these evidences, the present study aimed to explore the relationship between healthy HRM practices, healthy employees, and healthy organizational outcomes in a sample of small and medium enterprises' (SMEs) workers in the south of Italy.

The choice of this particular context was guided by the evidence that the local labor market where the study comprised mainly by SMEs. In view of this, results obtained in the study could contribute to the further investigation of HRM within this context. Moreover, the choice was based on the recognition of some distinctive traits within this kind of enterprise that could fit with the model proposed. According to the literature, SMEs are aware of the importance of considering their human resources as a top priority for developing and sustaining a competitive advantage [36]. In view of their specific work organization, which is relatively labor intensive in comparison with big enterprises, and because of their small size, each individual employee represents a substantial part of the workforce. Therefore, within SMEs, Human Resource Management is highly relevant.

Given these premises, the study assumed that SMEs could be conceived as HERO organizations, namely featured by three basic dimensions: healthy resources and practices, healthy employees, healthy outcomes. Nonetheless, given the specific context of the research, the study operationalized the dimensions partially following the original model [22] and proposing some integrations to their assessment.

As for "healthy resources and practices", following some traditional motivational approaches, such as Job Characteristics Theory [37] and Self-Determination Theory [38], the original model proposed to assess the job resources that allow employees to achieve work goals, stimulating their personal growth, learning and development. Moreover, organizations promote these job resources through healthy organizational practices addressed to increasing them both for employees and for the organization itself. Therefore, these resources were distinguished into task and interpersonal resources and organizational practices. Because of a previous validation study, the HERO model measured task resources through autonomy and feedback, interpersonal resources through climate,

coordination, teamworking, transformational leadership, and finally, organizational practices through self-constructed scales deriving from previous qualitative interviews.

In this study, task autonomy, communication, leadership, organizational mindfulness, and organizational commitment to resilience were considered privileged indicators of the organizational resources and practices and the antecedent of the model.

Compared with the original study almost all indicators for this dimension of the HERO organizations were considered, even if assessed through different measures. An integration was proposed referring to organizational practices that the present study investigated through organizational mindfulness and organizational commitment to resilience.

This choice was based on the analysis of further empirical evidences confirming the positive relationship between these variables, organizational efficacy, and employees' wellbeing. Yet, abundant research showed that employees' perception of autonomy in their job, a positive communication facilitating exchanges of information about tasks and processes, and an empowering leadership style could have positive impact on climate [39] and consequently on positive organizational behaviors [40,41].

Similarly, organizational mindfulness, meant as the ability of the organization to make sense of its practices and behaviors and to critically learn from experience, and organizational commitment to resilience were considered significant indicators of positive organizational outcomes [42,43].

In relation to the dimension referred to as "healthy employees' resources", the original model proposed to assess the cognitive and effective resources that employees invest in their work experience and that could be significantly influenced by the job resources afforded by the organization, consequently impacting on their performance. Namely, this dimension was measured through mental and emotional competencies, efficacy beliefs, trust, emotions, work engagement and resilience. With respect to this framework, the present study chose to focus attention only on work engagement, proposing the exploration of another crucial resource, psychological capital, that was proved to be significantly related to organizational practices and organizational outcomes.

Accordingly, in a recent meta-analysis published in 2011 considering a sample of 91 studies, Christian, Garza and Slaughter [44] highlighted the relevant role played by work engagement in the relationship between some features of the working context (e.g., autonomy, perceived support by colleagues, HRM practices, leadership styles) and task and contextual job performance. In the same direction, a special issue of the International Journal of Human Resource Management edited by Truss and colleagues in 2013 confirmed work engagement as a significant mediator in the relationship between HRM and performance [45–47].

In a similar vein, another meta-analysis by Avey, Reichard, Luthans and Mhatre [48] reviewing a sample of 51 independent groups of participants (a total of 12,567 workers) showed that psychological capital was significantly related to positive organizational behaviors, such as job satisfaction, commitment, psychological wellbeing, organizational citizenship, and performance, assessed both through of objective and subjective measures of efficacy.

Finally, the last dimension proposed by the HERO model was referred to as "healthy organizational outcomes". This dimension was originally measured through performance (in-role and extra-role behavior), commitment and results (excellence and service quality measured by customers).

In view of the smaller sample involved and of the different research design of the present study that did not considered multisource data (e.g., employees, supervisors, and customers) but focused only on employees' self-reported information, this dimension was simplified with respect to the original model considering extra-role behavior as the main indicator for "healthy organizational outcomes". The choice of a psycho–social subjective measure of performance was supported by further evidences confirming that this kind of behavior could be beneficial for organizations because they might be predictors of organizational wellbeing [10,49]. Moreover, the extra-role behaviors contribute to making the organization sustainable and to increase its competitive advantage.

*Aims of the Study*

Therefore, drawing from these assumptions, the main aim of the study was to explore the relationship between the "healthy resources and practices" (autonomy, communication, leadership, organizational mindfulness, and organizational commitment to resilience) as antecedent of the model, "healthy employees' resources" (work engagement and psychological capital) as mediators of the model, and "healthy organizational outcomes" (extra-role behavior) in a sample of SMEs workers in south of Italy. More specifically, the study attempted to propose the integration of some indicators for each of the three dimensions as shown by Figure 2.

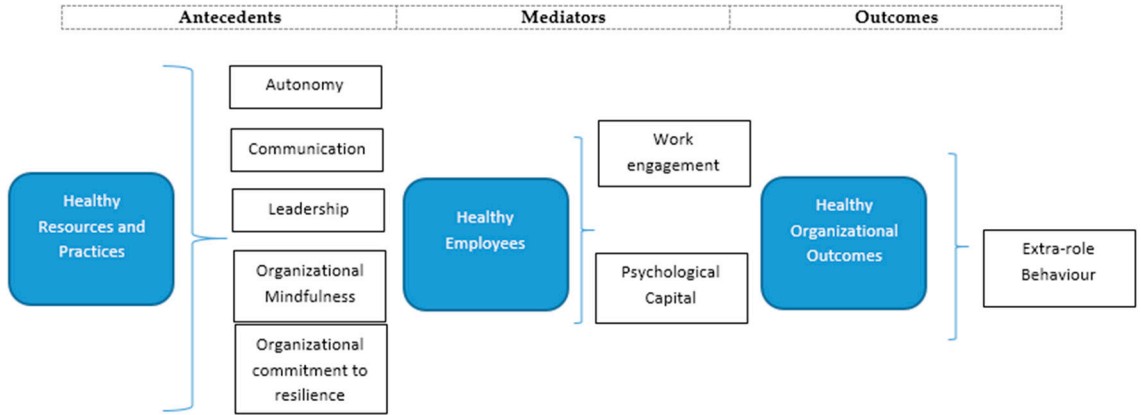

**Figure 2.** The research model.

The objective was to gain useful insights both for theory and HRM practices to contribute to the debate about the sustainable development of organizations [1], enhancing its intangible capital, and improving the wellbeing, satisfaction and performance of employees.

Accordingly, the study assumed that the relationship between the perception of the practices and of the organizational resources and extra-role behaviors was mediated by the individual resources of employees-work engagement and psychological capital as suggested by several empirical evidences in the field [12,50–52].

More specifically, the following hypotheses were formulated:

**H1:** *Work engagement and psychological capital mediates the relationship between autonomy and extra-role behaviors;*

**H2:** *Work engagement and psychological capital mediates the relationship between leadership and extra-role behaviors;*

**H3:** *Work engagement and psychological capital mediates the relationship between communication and extra-role behaviors;*

**H4:** *Work engagement and psychological capital mediates the relationship between organizational mindfulness and extra-role behaviors;*

**H5:** *Work engagement and psychological capital mediates the relationship between commitment to resilience and extra-role behaviors.*

## 2. Materials and Methods

### 2.1. Participants

The study involved 300 employees, working in a sample of heterogeneous small and medium enterprises located in Bari, in the south of Italy. Two hundred and thirty-six questionnaires were returned with a response rate of 78.6%.

Most of the participants involved (mean age 42.38, s.d. = 9.33) were males (89.84%) and only 10.16% females. As for education, 50.43% held a diploma, 31.77% completed middle school, and 17.80% received a degree. With respect to their tasks, 67.07% were workmen, 25% clerks, and a small percentage of managers (5.93%). As for the working sectors, participants were mainly employed in the production (69.91%), 15.25% worked in administration, 8.90% in Research and Development, 4.67% in purchases, and finally, the 1.27% in the commercial area.

### 2.2. Procedure

A convenience sample was involved through the administration of an online self-reported questionnaire. Participation was voluntary and anonymous. A cover letter was prepared to describe research aims and to guaranteed participants confidentiality. All participants were invited to fill in a form to give informed consent at the beginning of the online administration. The procedure was conducted according to the Italian data law on data protection (Legislative Decree No. 196/2003).

### 2.3. Measures

Data were collected using a structured questionnaire composed of a socio-anagraphical section, aimed at collecting information about gender, age, education, working role, and professional sector, and a second section, encompassing the measures chosen to assess the variables of the model described earlier.

All measures consisted of a series of assumptions assessing participants' agreement with each of them using a 5-point scale (1 = absolutely disagree) to 5 (completely agree).

Where available, Italian-validated versions of the scales have been adopted. In case of measures that were translated from the original version, the back-translation technique was used. Two mother tongue researchers independently translated the scales and then compared their versions by calculating an agreement index [53].

#### 2.3.1. Heathy Organizational Resources and Practices

Organizational resources were operationalized through measures of autonomy, communication, and leadership. Accordingly, to assess Organizational practices organizational mindfulness and commitment to organizational resilience were considered.

Autonomy, communication, and leadership were measured through the subscale of the same name encompassed in the Majer and D'Amato Organizational Questionnaire (M_DOQ10) [39]. More specifically, the "autonomy" subscale is composed of 6 items assessing the employees' perception about their personal initiative while accomplishing tasks (e.g., "In my job I feel I am quite autonomous"). Reliability for this subscale measured through the Cronbach alpha is 0.74. The "communication" subscale is composed of 12 items assessing the employees' perception about the availability of information and the efficacy of the communication networks within the organization (e.g., "Top down communication is simply to give instructions"). Reliability of this subscale is 0.82. Finally, the subscale "leadership" is composed of 8 items focused on the employees' perception about the management style (e.g., "When my responsible is present, I feel it is hard to say what I really think"). Reliability of this scale is 0.76.

Organizational mindfulness has been assessed through the Mindful Organizing scale elaborated by Vogus and Sutcliffe [54] and translated into Italian by Magnano, Platania, Ramaci, Santini and Di Nuovo [55]. It is composed of 9 items (e.g., "We talk about mistakes and ways to learn from them").

This scale assesses employees' perception about the practices that the organization uses to manage change and to display resilience toward unexpected events ($\alpha$ = 0.82).

Finally, commitment to resilience was assessed through the Commitment to Resilience scale elaborated by Weick and Sutcliffe [43] and composed of 10 items (es. "In this organization people know that they can trust each other"). The scale assesses the employees' perception about the organizational efficacy to cope with the external challenges and, thus, to develop resilience ($\alpha$ = 0.80).

### 2.3.2. Heathy Employees' Resources

Work engagement and psychological capital were considered to assess this dimension.

Work engagement was measured through the Utrecht Work Engagement Scale, elaborated by Schaufeli and Bakker in 2003 [56] and translated into Italian by Balducci, Fraccaroli and Schaufeli in 2010 [57]. The scale is composed of 9 items assessing work engagement meant as "a positive, fulfilling, work-related state of mind that is characterized by vigor, dedication, and absorption" [58]. These latter are the constitutive dimensions of the construct that refer respectively to energy and mental resilience while working, the willingness to invest effort in one's work, and persistence even in the face of difficulties ("In my work, I feel full of energy"), to involvement in one's work, and experiencing a sense of significance, enthusiasm, inspiration, pride, and challenge ("I am proud of the work I do") and to full concentration in one's work, whereby time passes quickly. Reliability of the scale measured through the Cronbach's alpha is 0.87.

Psychological capital was assessed through the PsyCap Questionnaire, elaborated by Luthans, Youssef and Avolio [59] and translated into Italian by Alessandri, Borgogni, Consiglio, and Mitidieri, [60]. This measure encompasses 24 items that concretely refer to a positive psychological state featured by optimism (e.g., "I believe that all the problems occurring at work always have a bright side"), hope (e.g., "I have several ways to accomplish the work goal"), self-efficacy (e.g., "I am confident in my performance that I can work under pressure and challenging circumstances") and resilience (e.g., "Although too much responsibility at work makes me awkward, I can get through to work successfully"). Reliability of the scale measured through the Cronbach's alpha is 0.86.

### 2.3.3. Heathy Organizational Outcomes

Extra-role behavior was considered to assess organizational outcomes. This variable was measured through the Extra-role behavior scale developed by Podsakoff, MacKensie, Moorman, and Setter in 1990 [61] and composed of 4 items. Extra-role behavior has been defined as a set of organizational citizenship behaviors manifesting the positive relationship between employees and the organization (e.g., "I get involved to benefit this organization"). More specifically, these are voluntary behaviors that concretely show engagement in practices going behind formal responsibilities and organizational demands to spontaneous support to the other members of the organization, exclusively for the sake of the organization itself ($\alpha$ = 0.79).

### 2.4. Data Analyses

To test the role of personal resources in the relationship between organizational resources and practices and organizational outcomes a parallel multiple mediator model was applied [62], using the PROCESS SPSS (Model 4) computational tool [63].

Given the limited sample size, and to prevent violation of normal distribution assumptions, the non-parametric bootstrapping method was used as a robust estimation of both direct and indirect effects [62]. Bootstrapping provided a confidence interval (CI) around the indirect effect of the independent variable (organizational resources and practices employees perception) on the dependent variable (extra-role behaviors) via the mediators (work engagement and psychological capital). Multiple mediations are significant if the interval between the upper limit (UL) and lower limit (LL) of a bootstrapped 95% CI do not contain zero, which means that the mediating effect is different from zero [2].

The present exploratory study aimed to investigate whether work engagement and psychological capital could mediate each component of organizational resources and practices, employee perception, autonomy (H1), leadership (H2), communication (H3), commitment to resilience (H4), and organizational mindfulness (H5), on extra-role behaviors.

Because a single questionnaire was used, common method variance was addressed following the indications found in the literature [64–68], specifically for protecting item consistency, social desirability, and reducing evaluation apprehension. Among all statistic methods, Harman's single factor test was used in this case. From the result, the Total Variance Explained of the first component accounts for less than 50% of the all variables in the model, and then the instrument is free from significant common method bias effects. Moreover, items were inserted randomly into the questionnaire and scales were graphically separated from each other.

## 3. Results

### 3.1. Descriptive Analyses

Before investigating the hypotheses of the study, some preliminary analyses were conducted.

Table 1 shows the distribution of mean scores and standard deviations for each variable and Pearson correlations between the constructs that were chosen to assess resources and organizational practices, employees' resources, and organizational outcomes of the model.

**Table 1.** Means, standard deviation, Cronbach alpha, correlations (N = 236).

| Variables | Mean (sd) | 1 | 2 | 3 | 4 | 5 | 6 | 7 | 8 |
|---|---|---|---|---|---|---|---|---|---|
| 1. Autonomy | 3.16 (0.81) | ($\alpha$ = 0.74) | | | | | | | |
| 2. Communications | 2.96 (0.74) | 0.306 ** | ($\alpha$ = 0.82) | | | | | | |
| 3. Leadership | 3.47 (0.76) | 0.337 ** | 0.484 ** | ($\alpha$ = 0.76) | | | | | |
| 4. Organizational Mindfulness | 3.12 (0.75) | 0.375 ** | 0.525 ** | 0.532 ** | ($\alpha$ = 0.82) | | | | |
| 5. Organizational commitment to resilience | 3.10 (0.66) | 0.337 ** | 0.582 ** | 0.303 ** | 0.676 ** | ($\alpha$ = 0.80) | | | |
| 6. Psychological Capital | 3.74 (0.49) | 0.501 ** | 0.349 ** | 0.380 ** | 0.379 ** | 0.308 ** | ($\alpha$ = 0.86) | | |
| 7. Work engagement | 3.78 (0.79) | 437 ** | 383 ** | 0.409 ** | 0.427 ** | 0.378 ** | 0.551 ** | ($\alpha$ = 0.87) | |
| 8. Extra-role Behaviors | 4 (0.78) | 0.426 ** | 0.177 ** | 0.123 | 146 * | 241 ** | 0.507 ** | 0.335 ** | ($\alpha$ = 0.79) |

\* $p < 0.001$ (2-tailed); \*\* $p < 0.005$ (2-tailed).

### 3.2. Mediation Analyses

Data analysis showed a partial confirmation of the hypotheses.

More in detail, with respect to H1, results highlighted a partial mediation of psychological capital in the relationship between autonomy and extra-role behaviors. Yet, autonomy was proved to be significantly related to extra-role behaviors. Though, by inserting psychological capital within the model, the relationship between these variables was amplified (Table 2).

These results are widely supported by abundant literature showing a direct positive relationship between task autonomy and organizational behaviors, such as motivation, satisfaction, and performance [69–72]. However, the evidence that psychological capital contributes to amplify this relationship provides precious information about the role of this personal resource, which might foster a positive elaboration of one's own ability to have control over and impact on the environment [73].

**Table 2.** Summary of multiple mediation analyses on autonomy and extra-role behavior (H1) (5000 bootstraps).

| Independent Variable | Mediators | Dependent Variable | Effect of IV on M | Effect of M on DV | Direct Effect | Indirect Effect | | Total Effect |
|---|---|---|---|---|---|---|---|---|
| IV | M | DV | (a) | (b) | (c′) | (a × b) | 95% CI | (c) |
| Autonomy | Work Engagement | Extra-Role | 0.41 (SE = 0.06) ** | 0.03 (SE = 0.07) n.s. | 0.21 (SE = 0.07) ** | 0.01 n.s. | (−0.059–0.083) | 0.041 (SE = 0.06) ** |
| | Psychological Capital | | 0.31 (SE = 0.04) ** | 0.50 (SE = 0.01) ** | | 0.18 ** | (0.105–0.286) | |

\* *p* < 0.001 (2-tailed); \*\* *p* < 0.005 (2-tailed).

On the other hand, there was no mediation effect of work engagement in the relationship between autonomy and extra-role behaviors, given the significant effect of autonomy on work engagement (Figure 3).

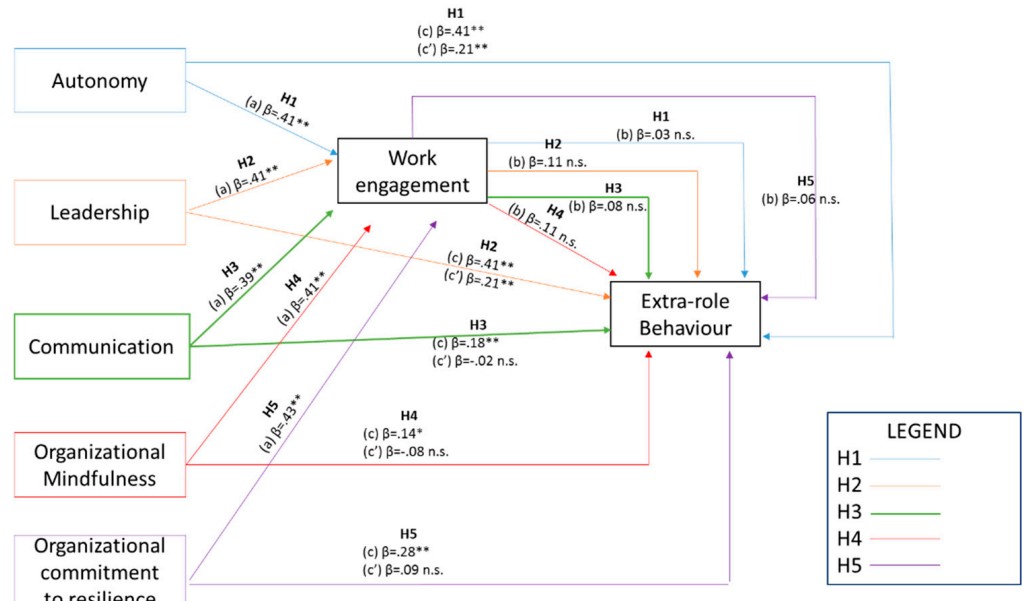

**Figure 3.** Relationships between organizational resources and practices and extra-role behavior with work engagement as a mediator.

There was no mediation neither of work engagement nor of psychological capital in the relationship between employees' leadership perceptions and extra-role behaviors (H2) (Table 3). These findings can be related to the sample composition. Accordingly, participants were employees coming from different organizational contexts. Therefore, being leadership perception a context-specific variable, linked to the peculiar features of the supervisor, this heterogeneity might have influenced the results.

**Table 3.** Summary of multiple mediation analyses on leadership and extra-role behavior (H2) (5000 bootstraps).

| Independent Variable | Mediators | Dependent Variable | Effect of IV on M | Effect of M on DV | Direct Effect | Indirect Effect | | Total Effect |
|---|---|---|---|---|---|---|---|---|
| IV | M | DV | (a) | (b) | (c′) | (a × b) | 95%CI | (c) |
| Leadership | Work Engagement | Extra-Role | 0.42 (SE = 0.06) ** | 0.11 (SE = 0.07) n.s. | −0.11 (SE = 0.07) n.s. | 0.05 n.s. | (−0.018–0.123) | 0.12 (SE = 0.07) n.s. |
| | Psychological Capital | | 0.25 (SE = 0.04) ** | 0.76 (SE = 0.11) * | | 0.19 ** | (0.113–0.308) | |

\* *p* < 0.001 (2-tailed); \*\* *p* < 0.005 (2-tailed).

Finally, there was a complete mediation of psychological capital in the relationship between communication (H3) (Table 4), organizational mindfulness (H4) (Table 5), organizational commitment to resilience (H5) (Table 6) and extra-role behaviors (Figure 4).

**Table 4.** Summary of multiple mediation analyses on communication and extra-role behavior (H3) (5000 bootstraps).

| Independent Variable | Mediators | Dependent Variable | Effect of IV on M | Effect of M on DV | Direct Effect | Indirect Effect | | Total Effect |
|---|---|---|---|---|---|---|---|---|
| IV | M | DV | (a) | (b) | (c′) | (a × b) | 95%CI | (c) |
| Communication | Work Engagement | Extra-Role | 0.39 (SE = 0.06) ** | 0.08 (SE = 0.07) n.s. | −0.02 (SE = 0.07) n.s. | 0.03 n.s. | (−0.028–0.105) | 0.18 (SE = 0.07) ** |
| | Psychological Capital | | 0.23 (SE = 0.04) ** | 0.73 (SE = 0.11) ** | | 0.17 ** | (0.089–0.276) | |

* *p* < 0.001 (2-tailed); ** *p* < 0.005 (2-tailed).

**Table 5.** Summary of multiple mediation analyses on organizational mindfulness and extra-role behavior (H4) (5000 bootstraps).

| Independent Variable | Mediators | Dependent Variable | Effect of IV on M | Effect of M on DV | Direct Effect | Indirect Effect | | Total Effect |
|---|---|---|---|---|---|---|---|---|
| IV | M | DV | (a) | (b) | (c′) | (a × b) | 95%CI | (c) |
| Mindfulness | Work Engagement | Extra-Role | 0.41 (SE = 0.06) ** | 0.11 (SE = 0.07) n.s. | −0.08 (SE = 0.06) n.s. | 0.04 n.s. | (−0.016–0.120) | 0.14 (SE = 0.07) * |
| | Psychological Capital | | 0.23 (SE = 0.04) ** | 0.75 (SE = 0.11) ** | | 0.18 ** | (0.096–0.276) | |

* *p* < 0.001 (2-tailed); ** *p* < 0.005 (2-tailed).

**Table 6.** Summary of multiple mediation analyses on commitment to resilience and extra-role behavior (H5) (5000 bootstraps).

| Independent Variable | Mediators | Dependent Variable | Effect of IV on M | Effect of M on DV | Direct Effect | Indirect Effect | | Total Effect |
|---|---|---|---|---|---|---|---|---|
| IV | M | DV | (a) | (b) | (c′) | (a × b) | 95% CI | (c) |
| Commitment to resilience | Work Engagement | Extra-Role | 0.43 (SE = 0.07) ** | 0.06 (SE = 0.07) n.s. | 0.09 (SE = 0.07) n.s. | 0.02 n.s. | (−0.040–0.103) | 0.28 (SE = 0.08) ** |
| | Psychological Capital | | 0.22 (SE = 0.05) ** | 0.71 (SE = 0.11) ** | | 0.16 ** | (0.077–0.271) | |

* *p* < 0.001 (2-tailed); ** *p* < 0.005 (2-tailed).

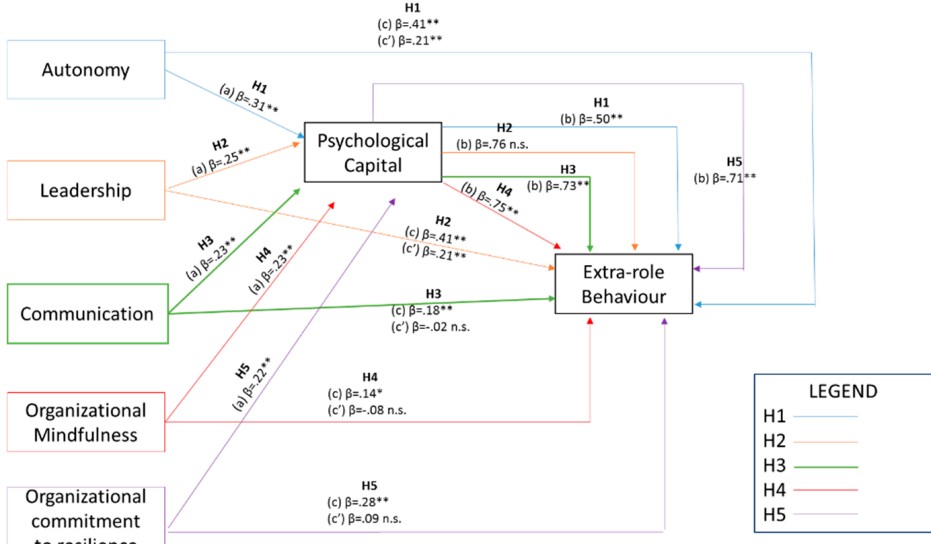

**Figure 4.** Relationships between organizational resources and practices and extra-role behavior with psychological capital as a mediator.

These results confirmed the precious role played by psychological capital in explaining positive organizational behaviors. Yet, as shown by many empirical evidences it is a personal resource that helps individuals achieve goals, buffering job demands and facilitating personal growth [74–81]. In this vein, research shows that human resource management (HRM) practices focused on training and development, employee feedback, information exchange and communication, employment security, innovation and learning, and positive coping with unexpected situations is associated with personal resources, such as self-efficacy, optimism, resilience and hope, improve employees work attitudes and behaviors, and in role and extra-role performance [82].

Therefore, from a primary prevention perspective addressed to support sustainable development [5,83–85], to invest in employees' personal resources is crucial to building organizational success and competitive advantage [86–91].

Conversely, there was no mediation effect of work engagement on the relationship between these variables although communication (H3), organizational mindfulness (H4), and commitment to organizational resilience (H5) were proved to be significant antecedents of work engagement.

These unexpected results seemed to converge on some recent studies showing that resourceful and flourishing work environments, namely contexts providing different job resources, such as feedback, social support, skill variety, might contribute to foster employees' work engagement [92–94], considering this construct preferably as an output of HRM practices rather than a mediator. Certainly, further research is needed to make more clarity about the role of such an important resource for human resources management and for sustainable development as well.

## 4. Discussion

Within the last few decades, healthy and resilient organizations have become a priority for academic research as well as for human resource management practices [34]. Despite this undebated focus, there are few empirical evidences that have investigated organizations from this point of view of the national context, which is characterized by an economic scenario mainly featured by Small and Medium Enterprises (SMEs), that is organizational systems that are peculiar and distinctive for dimensions, organizational praxis. and management styles.

Given this consideration, the explorative study presented above aimed to investigate the relationship between the perception of some HRM practices, employees' resources, and extra-role behaviors within a sample of the SMEs workers of the south of Italy.

Results partly confirmed the hypotheses of the study, paving the way for interesting future developments.

Accordingly, with reference to the role played by individual resources as mediators in the relationship between "organizational resources and practices" and "organizational outcomes", two meaningful aspects need attention: On the one side, the strategic function of psychological capital and on the other, the non-significant contribution of work engagement.

Yet, psychological capital was shown to be a significant mediator for most of the organizational resources and practices considered in the model: It partially mediated the relationship between autonomy and extra-role behaviors and fully mediated the relationship between the perception of organizational communication, the perception of living a mindful and resilient context and extra-role behaviors. Conversely, no mediation was found in relation to the perception of the leadership style.

Results followed the direction shown by the literature [9] and confirmed the strategic role played by psychological capital in the management of people, in the enhancement of sustainable performances [18] and of sustainable development of people and organizations [1–3]. The conclusion that seemed to emerge was that "healthy" organizational management practices, such as attention to communication, trust in collaborators, and a constant engagement toward a continuous improvement, could determine positive organizational behaviors only if sustained by the enhancement of people and of their resources, first their psychological capital. Therefore, this resource was proved to be fundamental in the management of the person/organization relationship, in guiding workers toward

the acknowledgement of the effort done by the organization to fulfil its mission and to keep their motivation high toward its objectives [95]. Therefore, employees' resilience, self-efficacy, optimism, and hope were proved to be precious resources that the organization should keep and nurture through focused HRM interventions.

Conversely, results of work engagement went in a different direction. Work engagement was proved to be a non-significant mediator in the relationship between organizational resources and practices and extra-role behaviors. Nonetheless, in support of this unexpected result, that disconfirmed even the empirical evidences coming from the original model [22] and a part of the hypotheses of the study, some more recent studies [96] underlined how the role of work engagement in relation to organizational outcomes is still debated and needs to be further investigated since it produced contradicting results. According to some scholars, work engagement could be considered an antecedent of positive organizational behaviors [97,98], while for others it could be a desirable outcome of good HRM practices [45,94].

Yet, although explorative in nature, since bound to a specific and limited organizational context (that of SMEs), results obtained in the study provide a contribution to this open debate about the relationship between HRM practices, employees' resources and employees' behaviors in the perspective of sustainable development [3].

Yet, if the psychology of sustainability and sustainable development suggest that well-being is a key issue for sustainable development and a fundamental requirement for organizational success and performance [3] then results obtained in the study seem to confirm the role of personal resources, such as employees' psychological capital in reinforcing the positive relationship between organizational practices and outcomes. Accordingly, healthy and resilient organizations are those contexts that being aware of this evidence engage themselves in the management of their human resources always keeping in mind the added value of people to their processes and goals.

## 5. Limitations, Implications for Future Research, Conclusions

The discussion of results allowed to highlight some limitations and, in the meantime, to sketch possible future research developments in the field.

A first limitation is related to the cross-sectional nature of the study and to the limited and heterogenous sample involved that did not allow to generalize results. Future research could plan longitudinal designs aimed to follow the same organization across time investigating if and to what extent specific organizational events (e.g., cultural changes, management transitions, economic pressures, vision, mission, and strategic policies) and specific configurations or the individual resources (e.g., in relation to age, role, seniority, occupational status, etc.) might influence the definition of organizational resources and practices and consequently have impact on extra-role behaviors.

A second limitation is related to the self-reporting measures used to assess the variables of the model. This option could have given a limited vision of extra-role behaviors as well as of HRM resources and practices. In addition to self-reporting measures, future research could also consider objective assessments of the same constructs (e.g., supervisors' assessments, indexes of participation to extra-role activities not defined by a formal job profile description). Another limitation related to the features of the tool used for the study could be the number of scales used, both in relation to the length of the questionnaire given to the workers and to the probability to find statistically significant relationships by chance. In this sense, some ex-post techniques to limit this assessment bias were adopted with reference to common method variance as reported earlier [64,65,67,68].

Moreover, the analysis of the correlation matrix between organizational resources and practices showed high values, such as in the case of the relationship between commitment to resilience and organizational mindfulness ($r = 0.676$) and between communication and commitment to resilience ($r = 0.582$). Consequently, another limitation could be found in the multicollinearity, that in turn suggested the hypothesis of content redundancy of some of the scales used. This limitation could be overcome by future research by simply eliminating some predictors or developing composite scores.

Finally, further organizational practices and/or individual resources could be inserted in the model to enrich the interpretation of organizational outcomes, that, as also shown by the present study, could be particularly dynamic and fluid.

With special reference to some of the possible implications highlighted by the study in terms of organizational practices, results showed that a working context designed to motivate employees to master their tasks and to trust their skills could be a determinant to encourage and support in-role and extra-role behaviors [99]. These assumptions confirm the significant amount of evidences found in the literature and reviewed in the study testifying the strategic role of HRM practices in determining the quality of performance and of the P/O relationship [48]. It follows that to adopt a human resource management style based on a constant engagement toward learning and toward the capitalization of knowledge is undoubtedly a competitive advantage for organizations if it becomes a strategy for organizational development. This option could generate positive effects on people and on their performance, motivating them to display spontaneous extra-role behaviors [100]. Therefore, it is the combination of the positive assessment that employees make of their organization, as a competent and resilient context ready to manage the unexpected, together with the awareness about one's own personal resources to determine extra-role behaviors, namely involvement and satisfaction [9,59–101].

In terms of strategic management, such awareness again confirms the crucial role that should be reserved to human capital within organizations, thus, further highlighting the need to foster a psychological perspective in the study of sustainability [1]. In this light, the HERO organizations are those that in addition to strategic planning of material and economic resources can catch and interpret employees' needs, can enhance skills, and capitalize resources. In a word, these are the organizations that can best aim at reaching sustainable performance [13] and sustainable development [2,3].

**Author Contributions:** The authors developed together the research project. A.M. drafted the theoretical section while M.L.G. worked on the data analysis. The discussion of results and the reviews of the original version were edited together as well.

**Funding:** This research received no external funding

**Conflicts of Interest:** The authors declare no conflict of interest.

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
