# Peer review of "People Make the Difference: An Explorative Study on the Relationship between Organizational Practices, Employees’ Resources, and Organizational Behavior Enhancing the Psychology of Sustainability and Sustainable Development"

_sustainability, doi:10.3390/su11051499_

Reviewer 1 Report

Thank you for considering me as a reviewer. 

I would suggest some changes that can improve the manuscript.

I think that the paper has very long Introduction and a lot of references. 

Instead of putting all the references at the end of the paragraph, like 8, 9, 10, 11, 12, 13, 14, 15, and 20, 21, 22, and 23, I would suggest you disperse them along the text. 

Related to your questionnaires, did you translate the PSYCAP into Italian? 

I think that you can better explain the procedure and the Ethical information for readers. 

Can you provide information about the specific Process Model used? It has been nº 4, I suppose. 

Some of the information in Table 1 seems that need adaptation to the Journal's template. 

Minor issues: References 26, and 29 are in another language than English. 

If you have used the model nº 4 from Hayes' software, perhaps you can add some figures to represent your results. Due that the Journal accepts color figures for free, you can include some of them because they are very informative for readers. 

Author Response

First of all we would like to thank our reviewers for the precious comments and suggestions given that we truly believe have greatly implemented the paper. We hope we could be considered for the next step of the revision. We have tried to answer to all the queries and marked in red in the paper any changes made. We attach the responses to all points raised.

Reviewer 2 Report

The paper respects the framework of the Special Issue onm psychology of sustainability.

The quality of presentations is high, as the researche design and the description of methods.

The presented research is interesting for scientific and professional communities.

Author Response

(The authors gave the same response as above.)

Reviewer 3 Report

First of all, I’d like to give my congratulations to the author/s for the manuscript and for giving me the opportunity to read it. Secondly,I have some concerns as outlined below. 

Theoretical development, objective and hypothesis

     More justification is needed about the relevance of sustainability and sustainable development area with the HERO model.

     More justification is needed about the expected relationships among healthy organizational resources and practices, extra role behaviors and the mediators (work engagement and psychological capital).

     Authors should specify in more detail the objective and the hypothesis of the study, which should be specifically formulated.

     Based on the HERO Model some aspects:

o  HERO Model is defined as HERO and not HE.R.O.

o  HERO model considers the collective nature of the measures but in the manuscript author/s are using the variables at individual level. An explanation about that will be useful. 

o  Some examples of variables in the manuscript are new and not included in original HERO Model. For example, authors talk about ‘material resources’, ‘motivation’ or ‘organizational mindfulness’ when explain the HERO Model. 

Participants and procedure 

     Taking into account the different organizations in the study, are there significant differences on the main variables based on the different companies? The same question is referred to the demographic variables. Are there significant differences on the main variables based on gender, education…?

     Please, give more information about the procedure in the study: for example, time spends to answer the questionnaires, how the questionnaires were administrated, the confidentiality or anonymity in the study and ethical aspects. 

Measures and Results

     Authors should to think about the possibility to test the results for method common variance.

     More explanation is needed about the selection of the variables in the study. Why extra role behavior, for example? Why organizational mindfulness? 

     Please, describe in detail which the different models tested are, the specific analyses made on and the results. 

     Which the explanation for the non-significant mediation of work engagement is? Why engagement has been measured as one dimension (and not by three).

     Which is the real novelty of the study for the HERO Model from the sustainability and sustainable development area?

     Which is the answer to the title question?

I hope that these previous comments are interesting for authors.

Author Response

First of all we would like to thank our reviewers for the precious comments and suggestions given that we truly believe have greatly implemented the paper. We hope we could be considered for the next step of the revision. We have tried to answer to all the queries and marked in red in the paper any changes made. We attach the responses to all points raised.

Round  2

Reviewer 3 Report

Generally, speaking the comments have been solved.